# The Suppression of Ubiquitin C-Terminal Hydrolase L1 Promotes the Transdifferentiation of Auditory Supporting Cells into Hair Cells by Regulating the mTOR Pathway

**DOI:** 10.3390/cells13090737

**Published:** 2024-04-24

**Authors:** Yeon Ju Kim, In Hye Jeong, Jung Ho Ha, Young Sun Kim, Siung Sung, Jeong Hun Jang, Yun-Hoon Choung

**Affiliations:** 1Department of Otolaryngology, Ajou University School of Medicine, Suwon 16499, Republic of Korea; yeonju0130@naver.com (Y.J.K.); jhflyflyfly@gmail.com (J.H.H.); jmyea@hanmail.net (Y.S.K.); jhj@ajou.ac.kr (J.H.J.); 2Department of Medical Sciences, Ajou University Graduate School of Medicine, Suwon 16499, Republic of Korea; inhee465@naver.com (I.H.J.); ylem2s1u@gmail.com (S.S.)

**Keywords:** UCHL1, auditory hair cells, supporting cells, transdifferentiation, mTOR pathway

## Abstract

In mammals, hearing loss is irreversible due to the lack of the regenerative capacity of the auditory epithelium. However, stem/progenitor cells in mammalian cochleae may be a therapeutic target for hearing regeneration. The ubiquitin proteasome system plays an important role in cochlear development and maintenance. In this study, we investigated the role of ubiquitin C-terminal hydrolase L1 (UCHL1) in the process of the transdifferentiation of auditory supporting cells (SCs) into hair cells (HCs). The expression of UCHL1 gradually decreased as HCs developed and was restricted to inner pillar cells and third-row Deiters’ cells between P2 and P7, suggesting that UCHL1-expressing cells are similar to the cells with Lgr5-positive progenitors. UCHL1 expression was decreased even under conditions in which supernumerary HCs were generated with a γ-secretase inhibitor and Wnt agonist. Moreover, the inhibition of UCHL1 by LDN-57444 led to an increase in HC numbers. Mechanistically, LDN-57444 increased mTOR complex 1 activity and allowed SCs to transdifferentiate into HCs. The suppression of UCHL1 induces the transdifferentiation of auditory SCs and progenitors into HCs by regulating the mTOR pathway.

## 1. Introduction

The loss of auditory hair cells (HCs) is the main reason for hearing loss [1], which is caused by a variety of factors, including aging, noise exposure, and ototoxic medications. In adult mammals, HC loss is irreversible, whereas in non-mammalian vertebrates, HCs can be regenerated from supporting cells (SCs). Neonatal mammalian cochleae also have HC regeneration capacity because they contain progenitor cells that can proliferate and differentiate into new HCs [2]. There are two modes of HC regeneration: direct conversion without division, a process termed direct transdifferentiation, and renewed SC proliferation by division and differentiation into HCs. The two key pathways regulating the stimulation of these endogenous progenitor cells are the canonical Wnt and Notch pathways [3]. The Wnt target gene leucine-rich repeat-containing G-protein receptor 5 (Lgr5) is a marker for HC progenitors. Lgr5^+^ progenitor cells can self-renew to regenerate HCs after HC loss in neonatal mouse cochleae in vivo. Lgr5 is expressed in third-row Deiters’ cells (3rd DCs), inner pillar cells (IPCs), inner phalangeal cells, and the lateral greater epithelium region [4].

Ubiquitin C-terminal hydrolase L1 (UCHL1; also known as protein gene product 9.5, PGP 9.5) belongs to the UCH family of deubiquitinating enzymes (DUBs). It is predominantly expressed in neurons, where it accounts for 1–5% of total soluble brain protein, but is also present in other tissues, such as the ovary, testis, and lung [5]. UCHL1 effectively hydrolyzes a peptide bond at the C-terminal glycine of ubiquitin to generate monomeric ubiquitin [6]. It acts as a free ubiquitin stabilizer, providing readily available ubiquitin for a variety of cellular events. The abnormal expression of UCHL1 is implicated in the pathogenesis of multiple diseases, such as nervous system diseases, cancer, and lung diseases [7]. Moreover, several lines of evidence suggest that UCHL1 is essential for the onset of cell genesis and differentiation and is a determinant of asymmetric distribution during germline stem cell self-renewal and differentiation [8].

The expression level of UCHL1 in the inner ear was unknown until recently. Our previous study showed that cochlear tissue expresses UCHL1 at levels as high as those in brain tissue, with the highest expression seen in spiral ganglion cells [9]. Additionally, we observed that, unlike adult cochleae, UCHL1 in the neonatal stage is expressed not only in neurons but also in SCs, similar to Lgr5. We hypothesized that the specific expression of UCHL1 in organ of Corti cells influences the postnatal production of HCs from SCs.

In this study, we investigated the spatiotemporal expression of UCHL1 during the development of mammalian cochlea and the potential role of UCHL1 in the transdifferentiation of SCs into HCs in neonatal cochleae.

## 2. Materials and Methods

### 2.1. Animals

Sprague–Dawley rats were obtained from Dae Han Bio Link Co., Ltd. (Chungbuk, Republic of Korea). All procedures involving animals were approved by the Institutional Animal Care and Use Committee (IACUC) of the Ajou University Graduate School of Medicine, Suwon, Republic of Korea (IACUC 2021–0023).

### 2.2. Organ of Corti Explants

After making an incision in the neck of Sprague–Dawley rats, the skin and mandibular bone were removed, and the skull was opened. Then, the brain was removed, and the temporal bone was collected in a sterile 60 mm petri dish containing phosphate-buffered saline (PBS). The lateral wall and spiral ganglion neurons (SGNs) were removed using fine forceps, and the organ of Corti was placed in a tissue culture plate and cultured overnight in Dulbecco’s modified Eagle’s medium (Gibco-BRL, Grand Island, NY, USA) supplemented with 10% fetal bovine serum (Gibco-BRL) and 0.06 mg/mL penicillin (Sigma-Aldrich, Steinheim, Germany) at 37 °C with 5% CO_2_.

### 2.3. Drug Treatment

After 24 h in culture, the cochlea was randomly divided into five groups. (1) The control group was cultured with dimethyl sulfoxide (DMSO) without any treatments. (2) We used 10 μM of the γ-secretase inhibitor, N-[N-(3,5-difluorophenacetyl)-L-alanyl]-(S)-phenylglycine t-butyl ester (DAPT; Calbiochem, Darmstadt, Germany) and 5 µM of the Wnt activator, CHIR-99021 (Sigma-Aldrich) to induce transdifferentiation of auditory SCs into ectopic HCs. (3) The explant was treated with 48 μM gentamicin (Sigma-Aldrich) for 24 h to induce damage to the HCs. (4) We checked if this induced loss of HCs could be prevented by the treatment of a specific UCHL1 inhibitor, LDN-57444 (5 μM, Selleckchem, Houston, TX, USA). (5) The explants that were treated with DAPT/CHIR ± LDN were additionally inserted with a mammalian target of rapamycin (mTOR) inhibitor, 2.5 nM rapamycin (Sigma-Aldrich) to identify the molecular mechanism of UCHL1. The culture medium was replaced once every 3 days after drug treatment. DAPT, CHIR-99021, LDN-57444, and rapamycin were dissolved in DMSO, and GM was dissolved in distilled water (D.W.). An equal volume of DMSO or D.W was used as the control.

### 2.4. Quantitative Polymerase Chain Reaction

Total cochlear RNA was extracted using RNAiso Plus (TaKaRa, Shiga, Japan), and cDNA was synthesized using a reverse transcription kit (TaKaRa) according to the manufacturer’s instructions. Quantitative PCR (qPCR) measurements were performed using the ABI Prism 7000 Sequence Detection System (Bio-Rad, Hercules, CA, USA) and the SYBR Green I qPCR Kit (NanoHelix, Daejeon, Republic of Korea), according to the manufacturer’s instructions. The qPCR primers were as follows: *Uchl1* (F) 5′- GAT TAA CCC CGA GAT GCT GA -3′, *Uchl1* (R) 5′- CTG AGC CCA GAG TCT CCT CC -3′; myosin VIIA (*Myo7a*) (F) 5′- GAG CTG CTG TGG CTG TGG ACA GGC C -3′, *Myo7a* (R) 5′- CAC CAG GTG TGG AGG GTA CTT C -3′; *Brn3c* (F) 5′- GTC TCA GCG ATG TGG AGT CA -3′, *Brn3c* (R) 5′- GCG ACA GGG TAA GAG ACT CG -3′; SRY-box 2 (*Sox2*) (F) 5′- GGG AAA TGG GGA GGG GTG CAA AAG AGG, *Sox2* (R) 5′- TTG CGT GAG TGT GGA TGG GAT TGG TG -3′; and *Gapdh* (F) 5′- AAC GAC CCC TTC ATT GAC C -3′, *Gapdh* (R) 5′- TCC ACG ACA TAC TCA GCA CC -3′. The relative quantification of gene expression was analyzed by the threshold cycle method, as described by the manufacturer’s manual (NanoHelix, Daejeon, Republic of Korea), and normalized to mouse *Gapdh* expression. The relative gene expression of target genes was calculated as 2-delta (Δ) Ct (ΔCt = Ct of target gene—ΔCt of control gene), and the fold change was calculated as 2 − ΔΔCt (ΔΔCt = ΔCt of target sample—ΔCt of reference sample). The expression of the gene of interest was expressed as fold change relative to the control group.

### 2.5. Immunohistochemistry

The dissected cochleae were fixed in 4% paraformaldehyde and stored overnight at 4 °C. Then, they were washed with PBS and decalcified in Calci-Clear Rapid Decalcifying Solution (National Diagnostics, Atlanta, GA, USA) for 3 days. For immunofluorescent histological analysis, the cochleae were embedded in paraffin using a specialized automated tissue processing system. The paraffin-sectioned slide was deparaffinized, rehydrated with grade alcohol, and processed for antigen retrieval using a citrate buffer (pH 6.0) at 95 °C for 15 min. After cooling, endogenous peroxidase was blocked in 3% hydrogen peroxide in methanol at room temperature for 10 min. The slide was blocked in 1% bovine serum albumin (BSA; GenDEPOT, Barker, TX, USA) for 1 h and then incubated overnight at 4 °C with the following primary antibodies: anti-UCHL1 (1:1000; #13179; Cell Signaling Technology [CST], Danvers, MA, USA), anti-SOX2 (1:1000; #4900; CST), and anti-MYO7A (1:1000; #25-6790; Proteus Biosciences, Ramona, CA, USA). After washing, the slide was incubated at room temperature for 1 h with fluorescein isothiocyanate (FITC)- or cyanine 3 (Cy3)-conjugated secondary antibodies. The nuclei were counterstained with 4′6,-diamidino-2-phenylindole (1:10,000; DAPI; Invitrogen, Paisley, UK). Images were taken with a Zeiss LSM 710 confocal microscope (Carl Zeiss Meditec, Jena, Germany).

### 2.6. Immunofluorescence

Cochlea explant tissues were fixed in 4% paraformaldehyde (Biosesang, Gyeonggi-do, Republic of Korea) for 15 min at room temperature, washed, permeabilized, and blocked in 1% BSA (GenDEPOT). Then, the cochlea tissues were incubated overnight at 4 °C with the following primary antibodies: anti-SOX2 (1:1000; CST), anti-MYO7A (1:1000; Proteus Biosciences), and anti-BRN3C (1:1000; sc-81980; Santa Cruz Biotechnology, Santa Cruz, CA, USA). The samples were thoroughly washed and incubated at room temperature for 1 h with FITC- or Cy3-conjugated secondary antibodies. The nuclei were counterstained with DAPI, and coverslips were mounted onto slides with a mounting medium (Vector Laboratories, Burlingame, CA, USA). The immunostained tissues were observed by confocal laser scanning microscopy (LSM 710; Carl Zeiss Meditec) at the Three-Dimensional Immune System Imaging Core Facility of Ajou University.

### 2.7. Western Blot Analysis

Cochlea explant tissue was lysed in a radioimmunoprecipitation buffer (25 mM Tris HCl, pH 8, 150 mM NaCl, 1% NP-40, 0.5% sodium deoxycholate, and 0.1% sodium dodecyl sulfate [SDS]) with a protease inhibitor cocktail (GenDEPOT). After extracting the proteins by centrifugation (13,000 rpm, 30 min), equal amounts of protein were loaded onto each lane of a gel, separated by SDS-polyacrylamide gel electrophoresis, and transferred to a polyvinylidene difluoride membrane (Millipore, Bedford, MA, USA) using electroblotting apparatus. The membranes were blocked in 5% skim milk for 1 h and then incubated with primary antibodies at 4 °C overnight. Then, the membranes were incubated with horseradish peroxidase-conjugated secondary antibodies (1:5000; GenDEPOT) for 1 h at room temperature. After washing the membranes, the proteins were detected using Enhanced Chemiluminescence Western Blotting Substrate (Thermo Scientific, Pierce Biotechnology, Waltham, MA, USA) and quantitated with densitometry analysis using ImageJ software 1.48v (National Institutes of Health, Bethesda, MD, USA). The following primary antibodies were used: anti-UCHL1 (1:1000; CST), anti-BRN3C (1:1000; Santa Cruz Biotechnology), anti-MYO7A (1:500; Proteus Biosciences), anti-SOX2 (1:500; CST), anti-phospho-P70S6 (p-P70S6) kinase (1:1000; #97596; CST), anti-P70S6 kinase (1:1000; sc-393967; Santa Cruz Biotechnology), and anti-β-actin (1:5000, #4970; CST).

### 2.8. 5-Ethynyl-2′-deoxyuridine Incorporation Assay

5-ethynyl-2′-deoxyuridine (EdU) is a nucleoside analog of thymidine incorporated into DNA during active DNA synthesis. Cell proliferation was determined by the EdU incorporation assay using the Click-iT^®^ EdU Imaging Kit (Invitrogen/Molecular Probes, Eugene, OR, USA) according to the manufacturer’s protocol. Briefly, tissues were treated with 10 µM EdU for 5 days and then fixed in 4% paraformaldehyde (Biosesang) for 15 min at room temperature. After washing with PBS and permeabilizing with 0.5% Triton X-100, the Click-iT^®^ reaction cocktail containing AlexaFluor^®^ 555-azide was added to the cells for 30 min. Then, immunofluorescence was performed for co-labeling with EdU and SOX2 or MYO7A.

### 2.9. Hematoxylin and Eosin Staining and Histological Analysis

Cochleae were harvested and fixed in 4% paraformaldehyde at 4 °C for 2 days. Then, they were washed with PBS and decalcified for 5 days using the Calci-Clear Rapid high-speed decalcifier (National Diagnostics, Atalanta, GA, USA). Decalcified cochleae tissues were embedded in paraffin using an automated tissue processing system. The paraffin blocks were sliced into 5 μm-thick sections. The slides were dried for 30 min on a slide warmer at 60 °C, deparaffinized, rehydrated in a graded alcohol series, and then stained with hematoxylin and eosin. The percentages of HCs, SGNs, and fibrocytes in the middle turn of the cochleae of at least five ears were scored, and the average was calculated.

### 2.10. Statistical Analyses

All values are expressed as the mean ± standard deviation. The groups were compared by the Mann–Whitney U test using SPSS software (version 12.0; SPSS Inc., Chicago, IL, USA). *p* < 0.05 was considered statistically significant. Sample sizes are indicated in the figure legends and were chosen arbitrarily with no inclusion and exclusion criteria. The investigators were not blind to the group allocation during the experiments and data analyses.

## 3. Results

### 3.1. Dynamic Expression of UCHL1 in the Developing Cochlea

Our previous RNA sequencing data showed that UCHL1 DUB is highly expressed in the postnatal day 3 (P3) organ of Corti. In addition, the cochlear and brain tissues showed a prominent expression of UCHL1 proteins. UCHL1 is also highly enriched in chicken vestibular HCs that regenerate, even after destruction [10]. First, we investigated the expression pattern of UCHL1 starting from embryonic day 17.5 (E17.5) to compare UCHL1 expression before and after the appearance of HCs. HC differentiation progresses in a basal to apical direction in the embryonic stage. At E17.5, UCHL1 was expressed in the lesser epithelial ridge (LER) corresponding to the future sensory epithelium and was notably stronger in the apical turn where HCs had not yet formed (Figure 1A,B). The expression of UCHL1 decreased from the apical to basal turns and was not expressed at the site where HCs developed (Figure 1B). In the basal turn, immature HCs with co-expressing HC and SC markers were generated, and UCHL1 was also co-localized with HCs (Figure 1B). Immunohistochemical analysis using the UCHL1 antibody shows that UCHL1 expression in the organ of Corti gradually decreased from E17.5 to P14 (Figure 2A). Interestingly, UCHL1 expression was restricted to 3rd DCs and IPCs from P2 to P7 (Figure 2A,C). At P9, UCHL1 expression completely disappeared in the organ of Corti cells and was, thereafter, expressed in nerve terminals and fibers connected to HCs from the SGNs (Figure 2A—red arrows). *Uchl1* mRNA expression also gradually decreased with the cochlear developmental stage (E17.5, P0, P2, P5, P9, and P14) (Figure 2B). The spatiotemporal expression of UCHL1 according to the age of birth from E17.5 is summarized in Figure 2C.

### 3.2. UCHL1 Expression Is Restricted to the 3rd DCs and IPCs in the Neonatal Organ of Corti

Regeneration studies in the neonatal in vivo inner ear are difficult due to the lack of an efficient in vitro system. To determine the role of UCHL1 in HC reprogramming, we set up an ex vivo system using a whole organ of Corti explant culture from P3 and performed an immunofluorescence assay to observe the UCHL1 expression pattern. We found that UCHL1 was highly expressed in 3rd DCs and IPCs (Figure 3). At P3, 1st and 2nd DCs were weaker than 3rd DCs and largely disappeared at P5 (Figure 2). In addition, we conducted co-immunostaining with neuron-specific class III β-tubulin to confirm that auditory nerve and UCHL1 expression was independent at P3 (Figure 3D). The expression of the UCHL1 of P3 rats is schematically drawn in Figure 3F.

### 3.3. A Subpopulation of UCHL1-Expressing Cells Switches to HCs under Transdifferentiation

Notch signaling inhibition with the γ-secretase inhibitor, DAPT, induces the transdifferentiation of SCs in neonatal cochlea. Furthermore, increasing Wnt signaling with CHIR-99021 not only promotes the proliferation of SCs but also extends the HC-inducible period in response to the inhibition of Notch signaling [11,12].

Notch inhibition and Wnt activation at P2 resulted in notable increases in MYO7A-expressing HCs and decreased the number of SOX2-expressing SCs in the outer HC region (Figure 4A–C). Co-labeling with MYO7A and SOX2 and a reduction in SOX2 showed that most SCs transdifferentiated directly into HCs (Figure 4B). qPCR also showed higher levels of *Brn3C*, the transcriptional nuclear gene in HCs, and lower levels of *Sox2* (Figure 4D). To examine cell division during HC transdifferentiation, we treated the organ of Corti with EdU throughout the experiments and observed some double-positive EdU/MYO7A and EdU/SOX2 cells after DAPT and CHIR-99021 treatment (Figure 4E–H). However, most of the supernumerary HCs were derived from direct transdifferentiation because there were few EdU-labeled MYO7A cells (Figure 4F). We also found that most of the EdU and cyclin D1 and SOX2 co-expressing cells were observed in the IPCs and 3rd DCs (Figure 4I) upon the induction of transdifferentiation. The functional links between the expression of UCHL1 and the mitotic transdifferentiation of SCs into HCs need to be elucidated. Consistent with the results obtained during cochlear HC development in vivo, a decreased level of UCHL1 was also found in 3rd DCs and IPCs on co-treatment with DAPT and CHIR-99021 at both the *Uchl1* mRNA and protein levels within the organ of Corti (Figure 5A,B). We found that UCHL1-positive cells were transdifferentiated into HCs when co-expressed with the HC markers BRN3C and phalloidin (Figure 5C,D).

### 3.4. The UCHL1 Inhibitor LDN-57444 Promotes the Transdifferentiation of SCs into HCs

To determine whether the inhibition of UCHL1 influences transdifferentiation, we cultured the P1 organ of Corti explants and treated them with LDN-57444, a reversible, competitive, active-site-directed inhibitor of UCHL1, alone or with DAPT and CHIR-99021, over the course of 5 days (Figure 6A). The combined treatment of DAPT and CHIR-99021 increased the number of HCs by approximately 1.5-fold in all organ of Corti turns, including the apical, middle, and basal turns, while no change with LDN-57444 itself was observed (Figure 6C,F). However, LDN-57444 significantly increased the *Brn3c* mRNA levels compared to the DMSO control under transdifferentiation conditions (Figure 6G). Next, we determined whether LDN-57444 could promote HC regeneration after damage by an ototoxic drug. Unlike cisplatin, aminoglycoside antibiotics enter sensory HCs via apical mechanoelectrical transduction channels and, ultimately, cause HC death. Since the SCs surrounding HCs are not affected by aminoglycoside antibiotics, the transdifferentiation of SCs into HCs is possible. As expected, there was no significant loss of SCs in the organ of Corti upon treatment with 45 µM gentamicin (Figure 6D). After 24 h of gentamicin treatment, the media was replaced for each group: vehicle (DMSO), DAPT/CHIR-99021, and DAPT/CHIR-99021 + LDN-57444, without gentamicin (Figure 6A). Gentamicin treatment resulted in a marked, progressive loss of HCs from the basal to apical turns of the organ of Corti. DAPT/CHIR-99021 treatment promoted HC regeneration in most turns (Figure 6E). LDN-57444 co-treatment was associated with significantly more HCs compared to the DAPT/CHIR-99021 treatment alone (Figure 6E,H). Consistent with HC counting, treatment with LDN-57444 further promoted *Brn3c* mRNA expression under the DAPT/CHIR-99021 treatment compared to untreated explants in response to gentamicin (Figure 6I).

### 3.5. mTOR Complex 1 Activation by a UCHL1 Inhibitor Promotes HC Transdifferentiation

Recent studies have demonstrated a link between UCHL1 and the mTOR pathway [13,14]. We found that decreased p-P70S6K is one of the downstream effectors of mTOR complex 1 (mTORC1) during DAPT/CHIR-99021-induced transdifferentiation (Figure 7A). Furthermore, the treatment with LDN-57444 under DAPT/CHIR-99021 significantly increased the levels of p-P70S6K. mTORC2-driven AKT also significantly increased under the LDN-57444 treatment (Figure 7A,B). The immunofluorescence of p-P70S6K in the organ of Corti cells was enhanced in the DAPT/CHIR + LDN groups compared to the DAPT/CHIR group (Figure 7C).

To determine whether mTOR activation was required for the potentiation of differentiation into HCs by LDN-57444, we used the mTORC1 complex inhibitor, rapamycin, to block mTOR signaling (Figure 7D–F). Previous studies have reported that a high dose of rapamycin induces HC death in the organ of Corti explants. In a concentration-dependent experiment with rapamycin, we identified 2.5 nM as the concentration that inhibited mTOR activity without loss of HCs. We found that co-treatment with 2.5 nM rapamycin reversed LDN-57444-mediated increases in both BRN3C protein and mRNA expression (Figure 7E,F). These results indicate that UCHL1 plays an important role in the transdifferentiation of SCs into hair cells by regulating the mTOR pathway.

## 4. Discussion

In this study, we found that UCHL1 expression was inversely correlated with the differentiation of HCs during cochlear development. In the P2–P5 organ of Corti, UCHL1 was expressed in the 3rd DCs and IPCs and was downregulated during HC transdifferentiation by Notch inhibition and Wnt activation. In particular, supernumerary HCs derived through division mainly arose from UCHL1-expressing 3rd DCs or IPCs. The inhibition of UCHL1 by LDN-57444 led to an increase in supernumerary HCs and activation of mTORC1, the effects of which were reversed by rapamycin treatment.

Understanding the molecular mechanisms that regulate HC reprogramming in non-mammalian and mammalian vertebrates is very important for the development of regenerative therapeutics. Ubiquitination and deubiquitination via the ubiquitin proteasome system regulate pluripotency and differentiation in stem and progenitor cells [15]. Recent studies have reported that DUBs are important for the maintenance of pluripotency in stem cells. The proteasome 26S subunit non-ATPase 14 (Psmd14) was identified as an important DUB of the 19S lid complex of the proteasome. Its activity is essential for stem cell maintenance. Psmd14 expression is high in mouse embryonic stem cells (ESCs), while its expression decreases during differentiation. Genetic silencing of Psmd14 induces ESC differentiation and a decrease in octamer-binding transcription factor 4 (Oct4), which is a master regulator of ESC pluripotency [16]. Loss of another class of DUB, ubiquitin-specific peptidase 3, in human embryonic carcinoma cells results in a decrease in protein levels of Oct4 [17]. Consistent with these findings, in this study, the high expression of UCHL1 in epithelial cells in the prosensory region at E17.5 gradually decreased as they differentiated into HCs. Even when transdifferentiation was induced by artificial methods, such as DAPT and CHIR-99021 injection, UCHL1 decreased as SCs differentiated into HCs. Lgr5 marks the inner ear progenitor cells, which also have the ability to regenerate new HCs via both mitotic regeneration and direct transdifferentiation [18]. The expression of Lgr5 also gradually decreased during the development and maturation of the organ of Corti [3]. These observations suggest that UCHL1 may be involved in HC progenitor proliferation and differentiation.

UCHL1 is essential for the onset of neurogenesis and the determination of asymmetric distribution during germline stem cell self-renewal and differentiation [14]. Regarding muscle regeneration, a recent study reported the upregulation of UCHL1 in denervated muscle after denervation injury. In that study, the knockdown of UCHL1 significantly accelerated myoblast differentiation for muscle regeneration and repair [19]. LDN 57,444 is a reversible, competitive, active-site-directed inhibitor of UCHL1. We examined the effect of UCHL1 using LDN-57444 and showed that more HCs developed under the transdifferentiation conditions, consistent with muscle regeneration. Under normal conditions, LDN-57444 did not show any numerical changes in HCs, and UCHL1 knockout mice exhibit no hearing phenotype as measured by auditory brainstem response (ABR) (https://www.mousephenotype.org/data/genes/MGI:103149, accessed on 13 April 2024). However, UCHL1 knockout mice display a phenotype of progressive paralysis, neurodegeneration, and osteoporosis [20,21]. Further laboratory and clinical studies are needed to determine whether UCHL1 inhibitors can be used for HC regeneration.

mTOR signaling is critical for tissue regeneration and repair processes, including axonal outgrowth, cell proliferation, and resident stem cell-based regeneration [22]. Shu et al. [23] reported that mTOR is not active in normal adult cochlea and that reactivation of the mTOR pathway together with *Notch1* may provide another route to reprogramming and HC regeneration. One recent study demonstrated that *Lin28b* plays an important role in the production of HCs by SCs in the mammalian cochlea in an mTORC1-dependent manner [24]. Consistent with reports regarding the mechanism by which other DUBs are involved in regeneration, the LDN-57444 treatment under transdifferentiation activated mTOR, which resulted in increased HCs. Furthermore, using the mTORC1 inhibitor rapamycin, we found that UCHL1 inhibitor-induced SC reprogramming required mTORC1-dependent signaling. Although the exact mechanism by which UCHL1 negatively regulates mTOR between cochlear SCs and HCs is unknown, the evidence suggests that UCHL1 destabilizes mTORC1 by antagonizing DDB1-CUL4 ubiquitin ligase complex-mediated ubiquitination of the raptor [25]. Consistently, in C2C12, a subclone of the C2 mouse myoblasts cell line, UCHL1 knockdown did not change the major proteins of the mTOR complex but decreased the protein turnover of PRAS40, an inhibitory factor of mTORC1, which, in turn, activated mTORC1 signaling [13]. Another study also found that UCHL1 was associated with the eIF4F translation initiation complex and promoted eIF4F assembly and protein synthesis despite its inhibitory effect on mTORC1. These results point to a new mechanism by which UCHL1 bypasses mTOR downstream effector 4E-binding protein 1 (4EBP1) to induce protein biosynthesis and demonstrate the important functional relationship between UCHL1 and MYC in lymphoma formation [14].

Interestingly, we found that dividing SCs are mainly 3rd DCs and IPCs associated with EdU incorporation and cyclin D1 labeling in the organ of Corti explants exposed to DAPT and CHIR-99021. This is likely related to the Wnt/β-catenin signaling associated with Lgr5 expression, which indicates cell division and differentiation capacity [26]; however, UCHL1 also upregulates the Wnt pathway via β-catenin stabilization and T-cell factor (TCF)-dependent transcription. UCHL1 has two TCF4-binding sites on the UCHL1 promoter [27]. Several studies have shown a direct relationship between cell division and UCHL1. Whether UCHL1 is an oncogene or tumor suppressor in various cancer types remains a subject of debate, but several studies have reported that tumorigenesis is linked to the high expression of UCHL1. In particular, UCHL1 can enhance the cyclin-dependent kinase activity implicated in the pathogenesis of neurodegenerative diseases [28]. Additionally, in uterine serous carcinoma, UCHL1 interacts with cyclin B1, which is essential for mitosis during tumor cell cycle progression. The treatment of ARK1 xenograft uterine serous carcinoma mice with the UCHL1 inhibitor LDN-57444 reduced tumor growth [29]. Further studies are needed to confirm the relationship between UCHL1 and cell division in the context of the production of ectopic HCs from the neonatal organ of Corti.

## 5. Conclusions

This study demonstrated for the first time the expression pattern of UCHL1 in the mammalian organ of Corti. The suppression of UCHL1 induces the transdifferentiation of auditory SCs and progenitors into HCs in the neonatal cochlea by regulating the AKT-mTOR-P70S6K-UCHL1-mediated pathway (Figure 8). Taken together, UCHL1 may be a novel regulator of HC regeneration.

## Figures and Tables

**Figure 1 cells-13-00737-f001:**
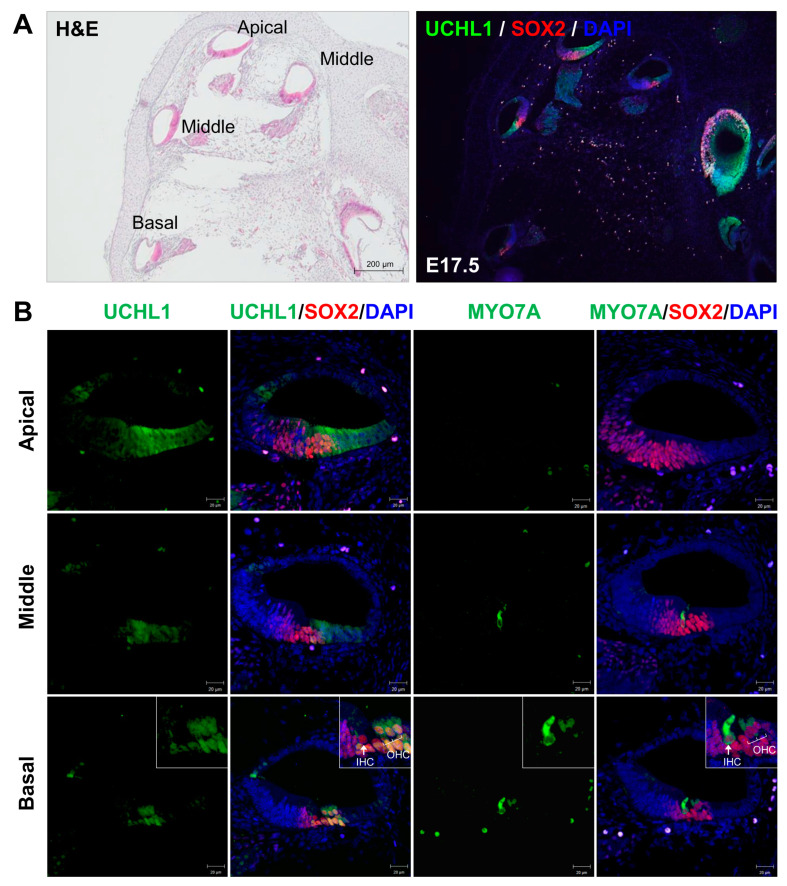
Expression of UCHL1 in E17.5 rat cochlea. (**A**) Low magnification H&E staining and immunohistochemistry of UCHL1 in cochleae on E17.5. Scale bar = 200 μm. (**B**) Representative immunohistochemistry images of UCHL1, SOX2, and MYO7A in the organ of Corti obtained from the apical, middle, and basal turns of the cochlea. Cell nuclei were counterstained with DAPI. The inset in the third row shows a magnified image of the organ of Corti. *n* = 3 each for immunohistochemistry. Scale bar = 20 μm. IHC, inner hair cell; OHC, outer hair cell.

**Figure 2 cells-13-00737-f002:**
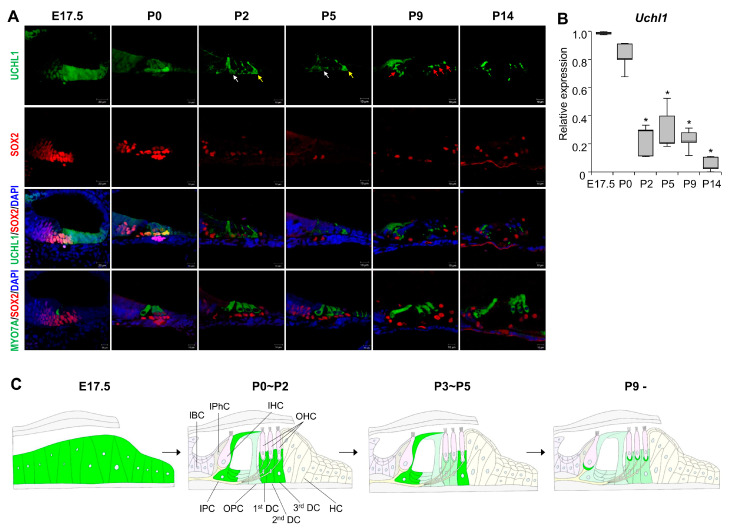
Immunohistochemical analysis of UCHL1 expression in the developing organ of Corti. (**A**) Representative images of UCHL1, SOX2, and MYO7A of the E17.5, P0, P2, P5, P9, and P14 rat organ of Corti obtained from the middle turn of the cochlea (*n* = 3 per group). White, yellow, and red arrows indicate IPC, 3rd DC and nerve terminal, respectively. Cell nuclei were counterstained with DAPI. Scale bar = 10 μm and 20 μm. (**B**) The expression levels of *Uchl1* mRNA in the cochlea from E17.5 to P14 were evaluated by qPCR. Relative expression levels of *Uchl1* plotted on the *Y*-axis after normalization to E17.5 samples (*n* = 3 per group) (* *p* < 0.05 by Mann–Whitney U test). (**C**) Schematic of the organ of Corti showing the UCHL1-positive (green) cell distribution at different development stages. IHC, inner hair cell; OHC, outer hair cell. IBC, inner border cell; IPhC, inner phalangeal cell; IPC, inner pillar cell; OPC, outer pillar cell; DC, Deiters’ cell; HC, Hensen’s cell.

**Figure 3 cells-13-00737-f003:**
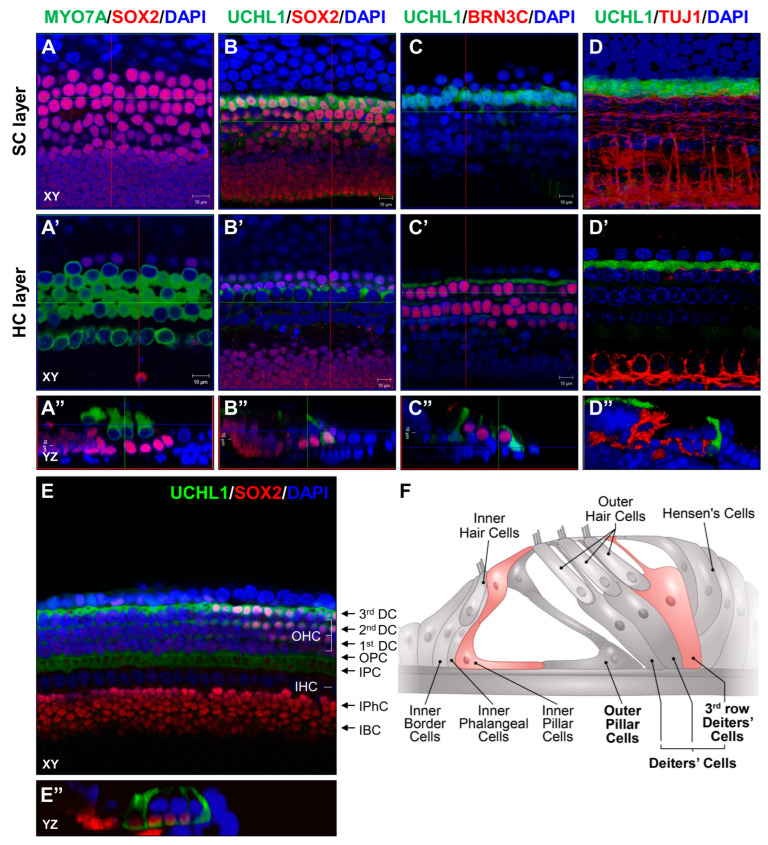
UCHL1 expression in a subset of SCs. Representative immunohistochemical staining of MYO7A/SOX2 (**A**), UCHL1/SOX2 (**B**), UCHL1/BRN3C (**C**), and UCHL1/TUJ1 (**D**) in the whole mount of P3 organ of Corti. (**A**–**D**,**A’**–**D’**) Confocal images were taken at the levels of SC nuclei and HC cytoplasm. (**A”**–**D”**) Three-dimensional orthogonal projections of a z-stack image. Scale bar = 10 μm. (**E**,**E’’**) UCHL1 immunostaining at low magnification in part of the organ of Corti. XY, confocal XY plane; YZ, confocal YZ plane. *n* = 3 each for immunohistochemistry. Scale bar = 20 mm. IHC, inner hair cell; OHC, outer hair cell. IBC, inner border cell; IPhC, inner phalangeal cell; IPC, inner pillar cell; OPC, outer pillar cell; DC, Deiters’ cell. (**F**) Illustration of organ of Corti showing UCHL1-positive SCs (red).

**Figure 4 cells-13-00737-f004:**
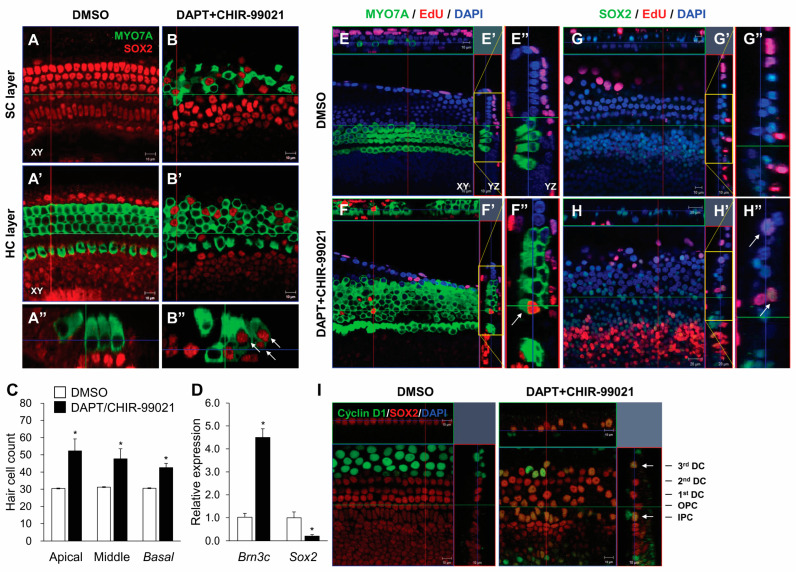
γ-secretase inhibitor and Wnt agonist-mediated induction of ectopic HCs. Representative immunohistochemical images of the apical turns of neonatal rat organ of Corti explants treated with DMSO (**A**) or 10 µM DAPT+ 5 µM CHIR-99021 (**B**) for 5 days. Samples were stained for HCs (MYO7A, green) and SCs (SOX2, red). (**A**,**B**,**A’**,**B’**) Confocal images were taken at the levels of SC nuclei and HC cytoplasm, respectively. (**A”**,**B”**) Three-dimensional orthogonal projections of a z-stack image. Scale bar = 10 μm. White arrows indicate cells positive for both antibodies. (**C**) Quantification of OHCs in DAPT/CHIR-99021-treated explants (for 5 days; *n* = 3 per group). (**D**) The expression levels of *Brn3c* and *Sox2* mRNA in the organ of Corti treated with DAPT/CHIR-99021 for 5 days were evaluated by qPCR. Relative expression levels of *Brn3c* and *Sox2* plotted on the *Y*-axis after normalization to DMSO control (*n* = 3 per group) (* *p* < 0.05 by Mann–Whitney U test). Immunohistochemical staining image for 10 µM EdU (red), with MYO7A (green) (**E**,**F**) and SOX2 (green) co-labeling (**G**,**H**). (**E**–**H**) Confocal images were taken of HC cytoplasm. Scale bar = 10 μm. (**E”**–**H”**) The yellow box indicates the magnified part of the z-stack image in (**E’**–**H’**). White arrows indicate EdU^+^/MYO7A^+^ or EdU^+^/SOX2^+^. (**I**) Immunohistochemical staining using anti-cyclin D1 (green) and SOX2 (red). The white arrows indicate co-localization of cyclin D1 and SOX2. IPC, inner pillar cell; OPC, outer pillar cell; DC, Deiters’ cell.

**Figure 5 cells-13-00737-f005:**
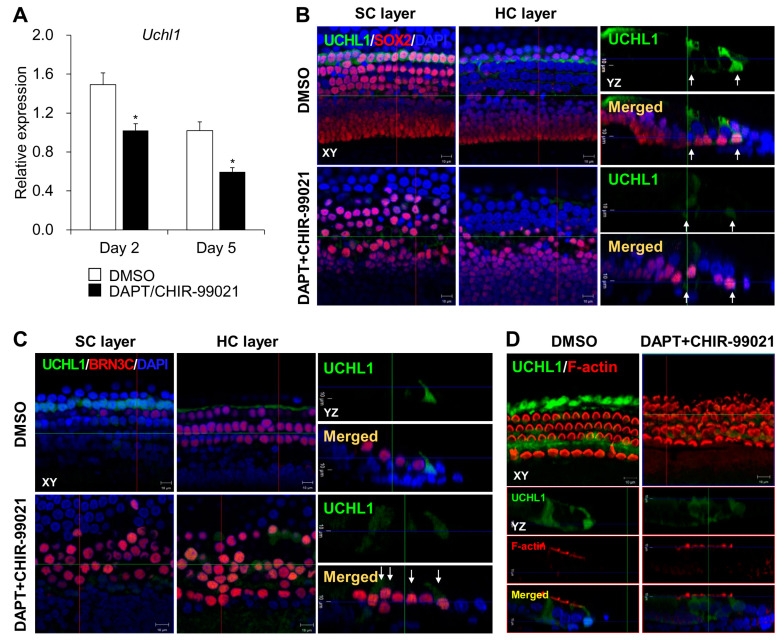
Expression of UCHL1 during transdifferentiation of SCs into HCs. (**A**) The expression levels of *Uchl1* mRNA in the organ of Corti treated with DAPT/CHIR-99021 from postnatal day 2 and 5 days were evaluated by qPCR. Relative expression levels of *Uchl1* plotted on the *Y*-axis after normalization to DMSO control (*n* = 3 per group) (* *p* < 0.05 by Mann–Whitney U test). (**B**) Representative images showing immunohistochemical staining of UCHL1/SOX2 in DMSO and DAPT/CHIR-99021-treated organ of Corti. White arrows indicate SCs expressing UCHL1. (**C**) Representative images showing immunohistochemical staining of UCHL1/BRN3C in DMSO and DAPT/CHIR-99021-treated organ of Corti. White arrows indicate SCs co-expressing UCHL1 and BRN3C. (**D**) Representative images showing immunohistochemical staining of UCHL1/F-actin in DMSO and DAPT/CHIR-99021-treated organ of Corti. *n* = 3 each for immunohistochemistry.

**Figure 6 cells-13-00737-f006:**
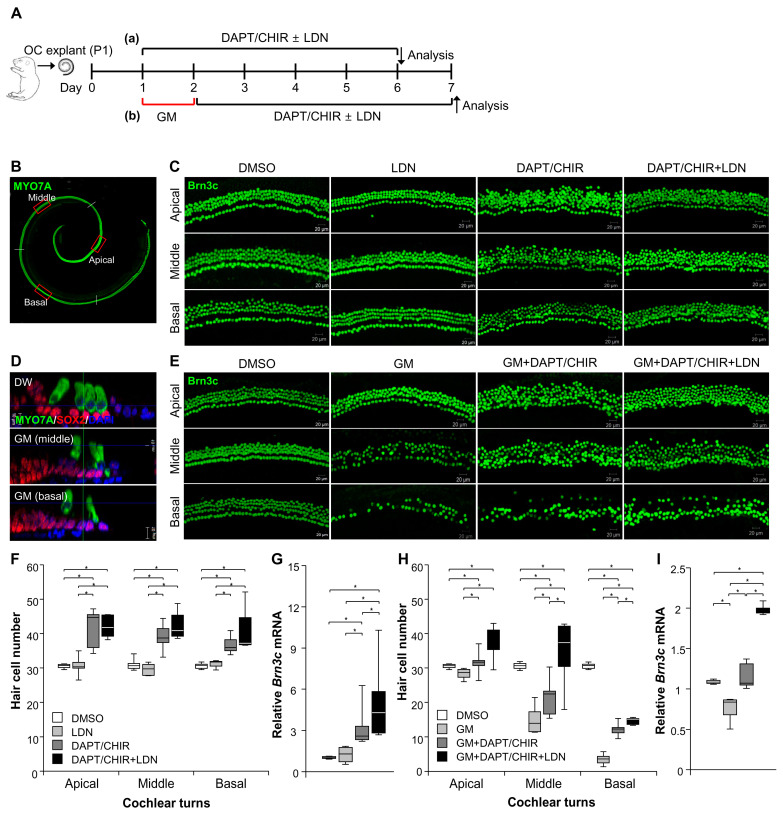
Pharmacological inhibition of UCHL1 by LDN-57444 increases the number of HCs. (**A**) Experimental timeline of ex vivo studies conducted under normal and ototoxic GM conditions to investigate the effects of LDN-57444 on transdifferentiation. (**B**) The cochlea was divided into three parts: apical, middle, and basal turns. Representative confocal images of BRN3C (green) in neonatal rat organ of Corti in treatment groups without (**C**) and with GM (**E**). Scale bar = 20 μm. Bar graph derived from quantitative analysis of BRN3C without (**F**) or with GM (**H**) at the three cochlear turns (apical, middle, and basal). (**D**) Confocal image showing that GM treatment destroys HCs but not SCs. Scale bar = 10 μm. The expression levels of *Brn3c* mRNA in treatment groups without (**G**) and with GM (**I**) were evaluated by qPCR. Relative expression levels of *Brn3c* plotted on the *Y*-axis after normalization to DMSO control (*n* = 5 per group) (* *p* < 0.05 by Mann–Whitney U test). OC, organ of Corti; CHIR, CHIR-99021; LDN, LDN-57444; GM, gentamicin.

**Figure 7 cells-13-00737-f007:**
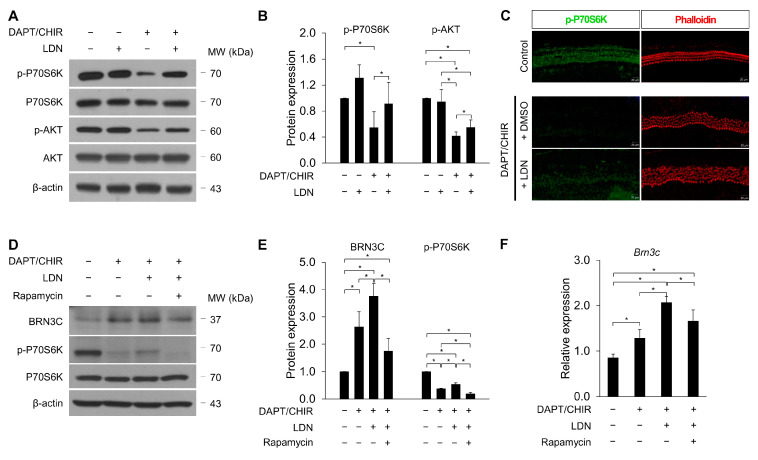
LDN-57444 enhances mTORC1 activity. Organ of Corti explants were treated with LDN-57444 along with DMSO or DAPT/CHIR-99021. Western blot (**A**) and densitometric analyses (**B**) of p-P70S6K and p-AKT. p-P70S6K and p-AKT levels were normalized to P70S6K and AKT, respectively. All bar graphs show mean ± standard deviation (*n* = 5 per group) (* *p* < 0.05 by Mann–Whitney U test). (**C**) Representative immunohistochemical staining of p-P70S6K (green) in organ of Corti explants with LDN-57444 under DAPT/CHIR-99021 treatment. Cell nuclei were counterstained with DAPI. *n* = 3 each for immunohistochemistry. Scale bar = 20 μm. Western blot (**D**) and densitometric analyses (**E**) of BRN3C p-P70S6K from organ of Corti lysates cultured in the absence or presence of 0.5 nM rapamycin with DAPT/CHIR-99021 for 5 days. (**F**) The expression levels of *Brn3c* mRNA in the different treatment groups were evaluated by qPCR. Relative expression levels of *Brn3c* plotted on the *Y*-axis after normalization to DMSO control (*n* = 5 per group) (* *p* < 0.05 by Mann–Whitney U test).

**Figure 8 cells-13-00737-f008:**
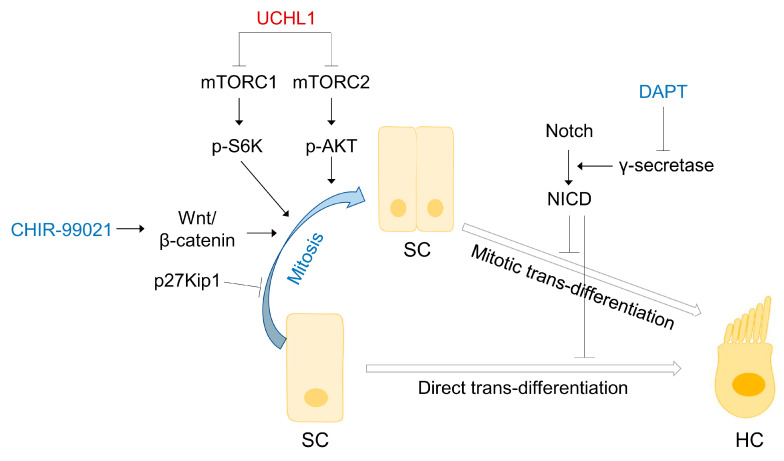
Schematic diagram depicting the negative involvement of UCHL1 in mTOR signaling in transdifferentiation of SCs to HCs.

## Data Availability

Data supporting the findings of this study are available within the article. The detailed data that support the findings of this study are available on request.

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
