# Peer review of "The Suppression of Ubiquitin C-Terminal Hydrolase L1 Promotes the Transdifferentiation of Auditory Supporting Cells into Hair Cells by Regulating the mTOR Pathway"

_cells, 2024, doi:10.3390/cells13090737_

Round 1

Reviewer 1 Report

Comments and Suggestions for Authors

This is an interesting study with some very focussed work on the role of Uchl1 in hair cell regeneration in rat cochlear explants. The confocal images are excellent and the overall approach is thorough. However, there are quite a few edits I would suggest making to the text in order to enhance the paper's approach and to clarify what has been done.

As a general note, the mouse knockout of Uchl1 exhibits no hearing phenotype as measured by ABR (https://www.mousephenotype.org/data/genes/MGI:103149). This does fit with some of the data (eg that LDN alone did not make a difference to the hair cell counts) but should be mentioned in the introduction or discussion since it is directly relevant. The knockout mouse does exhibit other phenotypes, which may be relevant to the use of Uchl1 inhibition to regenerate hair cells in humans.

L27-28 - Please reference the statement that the loss of HCs is the main reason for hearing loss. In my experience, hearing impairment is always seen before hair cell degeneration, suggesting that hair cell loss is secondary to whatever problem is causing the hearing impairment.
L36-38 - Please reference these three statements. Reference 3 describes ex vivo experiments and does not show where Lgr5 is expressed in the mouse inner ear.
L105 - Please reference or summarise the "threshold cycle method", since that name alone is not enough to describe the analysis.
L176-177 - Not all legends include the sample sizes, please check and correct (eg Fig 1, 3).
L183-5 - please reference these statements, and please mention which species is being discussed.
L204 - Again, it would be helpful to say "Expression of UCHL1 in E17.5 rat cochlea" - it's better if the reader doesn't have to page back to the methods to work out which species is being shown here.
Section 3.3 - this section is about investigating the role of UCHL1 in transdifferentiation, but there are two types of transdifferentiation, direct and mitotic, and it does matter which is which for this paper. However, in most uses of the word, it is not made clear which is meant. Please clarify. Eg does Notch inhibition with DAPT induce direct or mitotic transdifferentiation of SCs, or both? See also my comments about Fig 8, below.
L254 - it would be helpful to have the cell rows labelled on Fig 4I to make this point clearly.
L260 - what are "downregulated UCHL1-positive cells"? Cells which still express UCHL1 even though it's been downregulated? Please rephrase for clarification.
L268 - the white arrows are supposedly pointing to areas where Myo7a and Sox2 are colocalised, but there is no apparent colocalisation - there is no yellow overlap of green and red. If that's because Myo7a is a cytoplasmic stain and Sox2 is nuclear, it might be better to say "white arrows indicate cells positive for both antibodies" or similar.
Fig 5B - It would be helpful to have the white arrows on the merged panels as well as the UCHL1 panels.
L327-328 - This statement should be justified; "data not shown" should be avoided (as per the MDPI Cells guidelines, https://www.mdpi.com/journal/cells/instructions; also in L339).
Fig 7D - Is there a mistake with the lane labels? The first two are the same (all negative).
L399-400 - This sentence is very confusing. What is induced by DAPT/CHIR-99021 treatment, the mTOR activity or a reduction in the mTOR activity? What is enhanced by LDN-57444, the mTOR activity or a reduction in it? What promotes HC differentiation, the increase or decrease in mTOR activity? I suggest breaking this sentence up and rephrasing for clarity.
L411 - What is 4EBP1?
Fig 8 suggests that both DAPT and CHIR-99021 should be increasing mitotic transdifferentiation of SCs to HCs, but when they were applied to cochlear explants, most of the induced HCs were the result of direct transdifferentiation (L247, Fig 4B). This seems inconsistent.

Comments on the Quality of English Language

Some minor typos and grammatical errors. I have listed examples below, but this is not exhaustive, please do proofread after revisions:

L81 - extra "of" in the middle of the line, please delete.
L219 - "Dieters'" is a typo for "Deiters'".
L229 - extra "a" before "Figure", please delete.
L237 - "Dieters'" is a typo for "Deiters'", and on the figure, where it is spelled correctly, the apostrophe is in the incorrect place.
L342 - it should be "role in transdifferentiation", not "role of transdifferentiation".
L379 - erroneous line break
L432-433 - use either "through" or "by", but not "through by"
Fig 8 has the incorrect symbol for the gamma in gamma-secretase.

Reviewer 2 Report

Comments and Suggestions for Authors

Manuscript ID: Cells-2959573

Title: The Suppression of UCHL1 Promotes the Transdifferentiation of Auditory Supporting cells into Hair cells by Regulating mTOR Pathway

First Author: Yeon Ju Kim et al.

In this study, Kim YJ et al. aims to investigate the role of ubiquitin C-terminal hydrolase L1 (UCHL1) in the process of transdifferentiation of auditory supporting cells into hair cells. They studied the spatiotemporal expression of UCHL1 in rat cochlea during the development, and the potential role of UCHL1 in the transdifferentiation of auditory supporting cells into hair cells in ex vivo using neonatal whole organ of Corti explants. They demonstrated that the expression of UCHL1 gradually decreased as hair cells developed and was restricted to inner pillar cells and third-row Deiters’ cells between P2 and P7, suggesting that UCHL1-expressing cells are similar to the cells with Lgr5-positive progenitors. Immunofluorescence, quantitative PCR, and Western blot analyses in organ of Corti explants treated with UCHL1, γ-secretase, and mTOR inhibitors and Wnt agonist showed that UCHL1 expression was decreased even under conditions in which supernumerary hair cells were generated with a γ-secretase inhibitor and Wnt agonist. Moreover, the inhibition of UCHL1 led to an increase in hair cell number and activation of mTOR complex 1 activity that induced the transdifferentiation of auditory supporting cells and progenitors into hair cells.

The paper is well written, and the experiments are well designed and presented. The study is interesting and highlights a molecular mechanism involved in the regulation of auditory hair cell reprogramming by the ubiquitin proteosome system, which is important for the development of regenerative therapeutics.

Minor comments:

 - Figure 2: Please write the name of the different structures in the cryosection images. In the legend, add what each colored arrow indicates.

 - Page 9, line 277: Please replace “SIX2” with “SOX2”.

 - Page 14, line 431: Please replace “organ of Cortis” with “organs of Corti”.
